# Anemia in Breastfeeding Women and Its Impact on Offspring’s Health in Indonesia: A Narrative Review

**DOI:** 10.3390/nu16091285

**Published:** 2024-04-25

**Authors:** Ray Wagiu Basrowi, Andy Zulfiqqar, Nova Lidia Sitorus

**Affiliations:** 1Danone Specialized Nutrition Indonesia, Jakarta 12940, Indonesia; nova.sitorus@danone.com; 2Occupational Medicine Program, Department of Community Medicine, Faculty of Medicine, Universitas Indonesia, Jakarta 12300, Indonesia; 3Division of Urology, Department of Surgery, Faculty of Medicine, Universitas Gadjah Mada/Dr. Sardjito Hospital, Yogyakarta 55281, Indonesia; andyzulfiqqar@gmail.com

**Keywords:** anemia, iron deficiency, breastfeeding, child health

## Abstract

Anemia in breastfeeding women is a neglected global health issue with significant implications for maternal and child health. Despite its widespread occurrence and adverse effects, this problem remains largely unknown and overlooked on the global health agenda. Despite efforts to improve health access coverage and provide iron and folic acid supplementation, anemia persists. This underscores the need for a comprehensive approach to address the problem. Urgent action must be taken to prioritize education and awareness campaigns, ensure access to nutritious food, and enhance healthcare services. Education programs should focus on promoting iron-rich diets, dispelling cultural myths, and providing practical guidance. Improving healthcare services requires increasing availability, ensuring a consistent supply of iron supplements, and providing adequate training for healthcare providers. A successful implementation relies on a strong collaboration between the government, healthcare providers, and community. It is crucial that we acknowledge that high coverage alone is insufficient for solving the issue, emphasizing the importance of targeted interventions and a strategic implementation. By adopting a comprehensive approach and addressing the underlying causes of anemia, Indonesia can make significant progress in reducing its prevalence and improving the overall health of its population, particularly among breastfeeding women.

## 1. Introduction

Iron-deficiency anemia (IDA) is a pervasive public health issue, with its impact being particularly significant in developing countries like Indonesia [1]. This article focuses on the incidence of anemia in breastfeeding mothers and its impact on their offspring. During pregnancy and lactation, women face a higher risk of developing iron deficiency due to increased iron demands. This condition can have severe consequences for both the mother and her child, affecting maternal well-being, cognitive function, and the quality of breast milk, subsequently impacting infant growth and development.

In Indonesia, where breastfeeding is a widely practiced and culturally significant behavior, addressing IDA among breastfeeding mothers is of the utmost importance in order to ensure the health and well-being of both mothers and infants. Despite efforts to improve access to iron supplementation and healthcare services, the prevalence of IDA remains high. This underscores the need to explore the challenges associated with IDA in this specific population and identify effective interventions that can be implemented to alleviate the burden of anemia in breastfeeding mothers [2,3]. By understanding the unique factors contributing to IDA in Indonesia and implementing tailored strategies, we can strive towards better health outcomes for mothers and infants in the country, potentially serving as a model for addressing IDA in similar contexts worldwide.

## 2. Prevalence of Anemia in Breastfeeding Women: A Hidden Concern

The prevalence of anemia in breastfeeding women is an area that requires further exploration and understanding, as there are limited data available. The WHO estimates that, in 2019, the global prevalence of anemia was 29.9% in women of reproductive age. Prevalence rates were 29.6% in non-pregnant women of reproductive age and 36.5% in pregnant women [4]. Furthermore, the Indonesian National Health Survey (Riskesdas) revealed significantly higher prevalence rates among pregnant women (48.9%) in 2018 [5].

One study from an unspecified location reported a high prevalence rate of 60.3% in lactating women, with 20.3% of them experiencing severe anemia [6]. Other studies in Ethiopia revealed that the overall prevalence of anemia among breastfeeding mothers was 22.1% [7], while, in Indonesia, the prevalence is 52% [8]. This finding highlights the importance of investigating anemia specifically in breastfeeding women to determine the scope and severity of the issue. By delving into this unexplored landscape and conducting focused research on anemia prevalence, risk factors, and associated outcomes in breastfeeding women, clinicians and health authorities can gain valuable insights. It will enable us to develop targeted interventions and strategies to address the burden of anemia on maternal and infant health during the breastfeeding period.

## 3. The Root Causes of Anemia in Breastfeeding Women

Anemia is characterized by a lower-than-normal hemoglobin (Hb) concentration and/or red blood cell (RBC) count, insufficient to meet physiological needs. Hb concentration is the primary method for diagnosing anemia in both clinical and population settings. In terms of etiology, anemia can be divided into two types, which are nutritional anemia, and non-nutritional anemia (Figure 1). Nutritional anemia, largely due to inadequate iron intake, accounts for 42% of cases among children under five globally. Maternal, socioeconomic, and growth-related factors are linked to anemia in young children [7]. A study in Indonesia showed that socioeconomic status and education held a significant impact on maternal and children anemia. Poor nutritional status, old age, and a greater number of offspring were associated with a higher prevalence of maternal anemia. Low maternal education is associated with a higher prevalence of maternal and children anemia. Mothers with good education and knowledge tended to have a positive lifestyle and behaviors that can hinder maternal and child anemia, such as receiving iron supplementation and providing fortified milk for the offspring [9].

In developing countries, particularly Indonesia, iron-deficiency anemia is perceived as the most common cause of nutritional anemia, although some reports showed that iron-deficiency anemia contributes to less than 50% of nutritional anemia cases [10]. Addressing non-nutritional causes, such as malaria infection and RBC disorders (thalassemia and G6PD deficiency), is crucial for effective interventions and monitoring anemia control programs, particularly in malaria-endemic regions like Indonesia [8]. 

Nutritional anemias occur when essential nutrients crucial for red blood cell production or maintenance are insufficient, often stemming from causes such as inadequate dietary intake, heightened nutrient losses (e.g., due to parasitic infections, childbirth-related hemorrhage, or heavy menstrual bleeding), impaired absorption (e.g., lack of intrinsic factor for vitamin B12 absorption, high phytate intake, or *Helicobacter pylori* infection affecting iron absorption), or altered nutrient metabolism (e.g., deficiencies in vitamin A or riboflavin impacting iron store mobilization) [11]. While nutrient supplementation is a common strategy for preventing and treating nutritional anemias, the effectiveness of such supplements, particularly in terms of bioavailability and absorption, can vary, potentially limiting their impact [12].

Addressing anemia and iron deficiency is intrinsically linked to food security, especially in areas where limited access to nutritious food, as observed in food-insecure households, can lead to the insufficient intake of iron-rich foods, thereby increasing the risk of anemia [11]. In Indonesia, where only 8% of protein comes from animal sources, primarily fish, the high cost of animal protein in comparison to plant-based sources remains a concern. Despite an average daily per capita protein consumption of around 47.8%, with a significant portion from grains, there is a clear need to increase animal protein consumption, particularly to enhance iron intake and address nutritional deficiencies [13]. Low-income and food-insecure households encounter challenges in accessing nutrient-dense foods and healthcare services, exacerbating the impact of food insecurity on iron deficiency and anemia [14]. Furthermore, the comparison of the iron in milk, as shown in Table 1, reveals that Indonesia and India have lower iron levels in milk compared to developed countries with higher Global Food Security Index Scores that indicate better access to food security and economic purchasing power [15]. Table 1 shows the iron concentration of breast milk from women of term infants aged 1–6 months according to country.

## 4. The Unexplored Continuum Program for Anemia Management in Breastfeeding Women in Indonesia 

Breastfeeding iron supplementation is considered complementary rather than a primary focus. While antenatal care (ANC) visits and iron supplementation during pregnancy have received attention, the importance of iron supplementation during breastfeeding remains relatively underexplored. Meanwhile, recent reports have shown that the prevalence of anemia among breastfeeding women is significant [6], which can have adverse effects on both maternal and child health [8,24].

Inadequate knowledge regarding clinical indicators of anemia and limited support from husbands and families further hinder the prevention of anemia. Cultural taboos and traditional beliefs also impact dietary restrictions during pregnancy [25]. Overall, the inadequate supply of iron tablets, insufficient facilities and infrastructure, lack of counselling and guidance materials, and limited media information have all been factors contributing to the challenges faced in ensuring the receipt of iron tablets by anemic patients in various regions of Indonesia [26]. To address these challenges, several solutions can be implemented. Firstly, there is a need for the improved availability and accessibility of healthcare facilities, particularly in the eastern regions. Efforts should be made to ensure an adequate and consistent supply of iron tablets throughout the country. Additionally, healthcare providers should receive comprehensive training on anemia prevention and management, addressing competency gaps and ensuring consistent and accurate information is provided to the targeted population.

## 5. Closing the Loop: Integrating Anemia Management Programs into Breastfeeding Support Initiatives

The current postpartum anemia risk assessment algorithm supported by the CDC considers third-trimester anemia, excessive blood loss during delivery, and multiple births as risk markers [27]. However, previous studies suggest that additional factors such as multiparity, pre-pregnancy obesity, and not exclusively breastfeeding should be included [28]. These factors have been associated with an increased risk of postpartum anemia in previous studies. Multiparous women may be at a higher risk due to the inadequate recovery of iron status between pregnancies. Obesity may contribute to anemia through factors like greater blood loss and poorer diet quality [14]. Exclusive breastfeeding may protect against anemia by reducing iron loss or indicating a better socioeconomic status and dietary habits [20].

Despite the emphasis placed by organizations such as the WHO on the importance of monitoring the nutritional status of lactating women, this practice is rarely implemented [29] Micronutrient deficiencies in lactating mothers can lead to a significant decrease in the concentration of these nutrients in breast milk, ultimately resulting in infant depletion. Prioritized nutrients for lactating women include thiamin, riboflavin, vitamins B-6 and B-12, vitamin A, and iodine, as low maternal intake or stores can diminish the levels of these nutrients in breast milk, but maternal supplementation can help alleviate this issue. One study in Indonesia revealed that the majority of exclusively breastfed infants and their mothers had micronutrient intakes below recommended levels. Additionally, associations were found between maternal intakes and breast milk concentrations for three nutrients. The median daily infant intakes of iron, zinc, selenium, magnesium, sodium, and B-vitamins (thiamin, riboflavin, niacin, pantothenic acid, B-6, and B-12) were observed to be below their respective adequate intake [16,30,31]. This serves as a stem of anemia in breastfeeding women, particularly in Indonesia, which impacts their offspring. One study in India showed that infants whose mothers received iron supplementation for 6 months showed significantly higher levels of hemoglobin and serum ferritin compared to those whose mothers received iron for 3 months. It indicates that the maintenance of iron status and prevention of anemia in almost three-fourths of the infants at 6 months of age without iron supplementation may be attributed to the higher iron content of breast milk [32]. Thus, with regard to the commitment to ensure a continuum of care beyond the lactation period, it is crucial that we integrate anemia management programs into breastfeeding support initiatives. This integration will allow for ongoing screening, appropriate iron supplementation, and the continuous monitoring of anemia during the entire reproductive journey. 

In terms of screening and management, the Indonesian program lines up with the CDC and WHO recommendations, which is that universal screening for iron-deficiency anemia in pregnant women is crucial, with universal iron supplementation unless certain genetic disorders are present. The recommended daily dietary allowance during pregnancy is 27 mg, while lactation requires 9 mg, yet a typical diet only provides around 15 mg of daily iron. The CDC and ACOG advise screening for iron-deficiency anemia with a complete blood count (CBC) twice during pregnancy: in the first trimester and between 24 weeks and 28 weeks and 6 days [33,34]. This screening may include evaluating RBC indices, serum iron levels, and ferritin. Empiric treatment with iron is considered reasonable, with an expected improvement in hemoglobin and hematocrit levels. while the 2020 Indonesian Integrated Antenatal Guidelines recommend iron and folic acid tablets for pregnant women with anemia, along with efforts to prevent and treat malaria in malaria-endemic areas and appropriate medication for helminth infections, and screening for TBC and other non-nutritional causes of anemia [8,17].

The constraints of program implantation are reported in a study that reported anemia screening rates were low in primary healthcare facilities, with only a small percentage of pregnant women (28%) and breastfeeding women being screened for anemia. The researchers identified various factors that hindered the implementation of anemia screening, including the absence of standardized protocols, limited availability of screening tools, inadequate training of healthcare providers, and challenges in the referral system [35]. Based on their findings, it is concluded that improvements are needed in the anemia screening program for pregnant and breastfeeding women in primary healthcare facilities in Indonesia. Addressing the gap between national and international guidelines and the current decentralized approach to antenatal care requires health system solutions that support testing at integrated health center and primary health center levels. Simple rapid diagnostic tests for anemia at the point-of-care could be implemented at the integrated health center, increasing coverage and feasibility [36]. These options are offered in Indonesian guidelines as a means to improve screening rates.

## 6. The Hidden Consequences: How Maternal Anemia during Breastfeeding Impacts Children’s Health 

Anemia in lactating mothers is an often-overlooked public health issue in Indonesia, with far-reaching consequences for both the mother and the newborn [37]. This condition is closely linked to anemia experienced during pregnancy, as the susceptibility to iron depletion persists during lactation. The combination of iron depletion, coupled with the detrimental effects of blood loss during childbirth, significantly increases the risk of anemia in lactating mothers [38].

The association between postpartum anemia and an insufficient milk supply poses a significant challenge to successful breastfeeding and overall maternal–infant health [38]. Lactating women are also particularly vulnerable to anemia, as their iron stores are depleted to maintain the iron quality in breast milk, especially when their energy and iron intake are inadequate. This can have an impact on infants, as those born to anemic mothers have lower levels of hemoglobin (Hb) and ferritin compared to those born to nonanemic mothers [39]. However, it is important to note that the iron content in breast milk is optimal for the development of infants [39]. During the first six months of life, exclusively breastfed infants rely on their iron stores, since human milk has minimal iron content [40].

Another study’s findings in Indonesia suggest that human milk mineral concentrations were not notably affected by milk volume, apart from β-carotene, α-carotene, and β-cryptoxanthin, which exhibited a modest increase with greater milk output [27]. Notably, interactions between time and milk volume were observed, particularly at 5 months postpartum, indicating a stronger relation between milk volume and certain minerals such as calcium, selenium, TMP, riboflavin, and FMN. Conversely, zinc demonstrated a stronger association at 2 months postpartum [27]. Moreover, maternal iron biomarkers and hemoglobin were linked to milk iron concentrations at 5 months postpartum, even after adjusting for several factors including parity, human milk volume, rural living status, and inflammation [27]. Positive associations were also observed between maternal biomarkers such as RBP, selenium, and vitamin B-12 concentrations in serum and the corresponding milk concentrations of retinol, β-cryptoxanthin, selenium, and vitamin B-12, respectively [27]. These findings emphasize the significant impact of maternal biomarkers on milk mineral concentrations, with implications for the potential benefits of iron, zinc, and calcium supplementation during the postpartum period. Additionally, maternal hemoglobin levels affect the immunological and nutritional composition of breast milk throughout various maturation stages, emphasizing the need for targeted interventions to support anemic mothers in providing high-quality milk for their infants [28].

Two studies in Indonesia explore the correlation between anemia in breastfeeding mothers and anemia in infants, suggesting that the iron reserves in newborns are independent of breast milk. Anemic breastfeeding mothers were also more likely to have infants with stunting. However, the association between anemia and stunting in infants aged 0–6 months was not significant [8]. Two studies in Indonesia explore the correlation between anemia in breastfeeding mothers and growth in infants. These studies revealed conflicting results. Studies by Sudaryati et al. revealed that anemia in lactating mothers is not associated with infant growth and nutritional status [41]. However, Rini et al. [42] found that anemia during lactation had a significant low to moderate correlation with infant weight [42]. Studies examining the relationship between these two variables were scarce. Furthermore, vitamin D deficiency might play a role in the development of stunting as Indonesian women have a high prevalence of vitamin D deficiency [43]. Kumar et al. studied the breast milk iron content of anemic mothers and found that the breast milk iron content is reduced significantly only in severely anemic mothers [40]. Several studies revealed that maternal hemoglobin concentration during breastfeeding was associated with infants’ hemoglobin concentration [37,44,45,46]. On the other hand, Marques et al. revealed no significant association between maternal anemia and their children’s anemia [44]. A systematic review and meta-analysis in Ethiopia showed that anemia is related to childhood stunting [47]. Together, most of these findings pointed toward the association between anemia during lactation and infants’ anemia and child growth (Table 2). 

Another study conducted by Luo et al. revealed that infants between 6–11 months of age experienced a sharp drop in hemoglobin levels at around 6–7 months, followed by a slower decline at 8 months, and a subsequent increase until 11 months of age. Infants who were still breastfed more than 6 months of age had lower hemoglobin concentration and a higher prevalence of anemia compared to children who had previously received formula milk [49]. This study highlights the importance of addressing anemia in breastfeeding mothers which could lead to a lower iron content in breastmilk and higher risk of anemia in later infancy period. 

Iron plays a vital role in the rapid development of the central nervous system during infancy, impacting aspects such as myelination, neurotransmitter function, and energy metabolism. Studies suggest that early iron deficiency may be associated with long-term neurodevelopmental impairments [50]. Infants rely on iron for red blood cell production, primarily drawing on stores acquired in the last gestational months. By 4–6 months of age, these reserves can become depleted, particularly in cases of low birth weight, prematurity, rapid growth, insufficient dietary iron intake, early introduction of low-iron complementary foods, prolonged milk feeding, and intestinal parasitic infections causing blood loss. In the preschool years, rapid growth increases iron requirements, while, in the third year, growth slows, and children are more susceptible to intestinal parasitic infections and inadequate diets. School-age children with iron deficiency may experience impaired cognitive and physical development, though a definitive causal link is not confirmed. Ensuring optimal nutrition in school-age children could have broader benefits beyond academic performance [51].

Tackling the nutrition issue in mothers is crucial for the long-term growth and development of their children. Numerous studies have highlighted the significant impact of maternal nutrition on children’s outcomes, emphasizing the need to address maternal nutrition for optimal results later in life [25]. Improving the dietary intake and overall nutritional status of mothers not only positively influences their children’s growth and development but also plays a vital role in reducing the risk of health and nutrition deficiencies. Anemia, in particular, should be addressed in lactating women as it can have detrimental effects on both the mother and child, including reduced milk production, postpartum depression, compromised immunity, and impaired cognitive development. Enhancing blood levels and iron reserves in lactating women contributes to global initiatives aimed at reducing maternal mortality and ensuring the well-being of both mothers and children. Comprehensive nutrition interventions that target both mothers and children are essential in achieving these goals.

Complimentary feeding (CF) is also an important factor in breastfed offspring. The timing of introducing the CF should be optimal as a delayed introduction is associated with a low height-to-age Z-score. The recommendation is to start CF prior to seven months old, especially in developing countries [52]. Another study also showed the beneficial effects of early CF introduction. The introduction of CF prior to 4 months old was associated with an increase in children’s body mass index (BMI) [53].

Notably, several confounding factors should be taken into account in deciphering the impact of anemia in breastfeeding women on the offspring. Breast milk volume and frequency could become a confounding factor since most studies did not measure this variable [41]. Environmental factors could also influence the infant’s outcomes since a dirty environment can lead to a sick child or infection which can induce anemia [42,44]. Inherited blood disorders should also be investigated first to rule out other causes of anemia in children [45]. Studies in Brazil also stated that the clamping time of the newborn’s umbilical cord can influence the anemic status since early clamping is associated with a higher incidence of anemia [44,48]. The newborn’s condition is also important in predicting the outcomes as an inadequate birthweight is associated with the risk of anemia [37].

## 7. Unraveling the Challenges and Charting the Future of Supplementation in Indonesia

Anemia is caused by multiple factors and is a complex issue. The insufficient intake of iron-rich foods has often been identified as a primary cause. However, the success of an anemia control program is influenced by various factors, including the government’s commitment to addressing anemia, the capacity of healthcare providers, food availability, deficiencies in other micronutrients that contribute to red blood cell production, and the motivation of patients themselves.

In Indonesia, providing iron and folic acid supplementation is an efficient method for preventing anemia resulting from iron and/or folic acid deficiency. These blood-booster tablets (TTD) are administered to women of childbearing age and pregnant women, with pregnant women advised to take them daily throughout their pregnancy or a minimum of 90 tablets, under the Ministry of Health guidelines from 2014 [6]. 

Several studies have explored the impact of maternal iron supplementation on breastfeeding infants and the composition of breast milk, yielding varied results that underscore the complexity of iron metabolism and its interaction with other minerals. While short-term studies have shown limited effects on iron status, longer-term studies of up to 6 months have demonstrated the potential benefits of supplementation [54,55]. However, it is crucial to view iron supplementation not as a quick fix for anemia but as an ongoing relay program that requires consistent and sustained efforts. 

In terms of the impact on breast milk composition and infant iron status, research has shown that longer follow-up periods are needed to fully understand the effects of maternal iron supplementation during lactation. While some studies found no significant differences in serum iron and ferritin levels, others demonstrated a rise in serum ferritin and hemoglobin levels, particularly in initially anemic infants, when medicinal iron supplementation was introduced during complementary feeding [54,55]. Interestingly, breast milk iron levels declined over time regardless of maternal anemia status, suggesting independent regulation to prioritize infants’ needs. The interaction with other minerals such as copper remains complex, with iron supplementation decreasing serum copper concentrations and inconclusive associations with zinc and copper status [56]. 

Despite the economic growth and improvements in living standards in Indonesia, the prevalence of anemia and iron deficiency among women of reproductive age remains high [57,58]. This indicates that economic status alone is not sufficient to address the issue of anemia. While economic growth provides individuals with greater purchasing power, enabling them to afford a more diverse and nutritious diet, it does not guarantee knowledge, access to, and consumption of iron-rich foods necessary for preventing anemia [59,60]. This highlights the urgent need for education and campaign programs to enhance understanding and address these issues within the population [59].

## 8. Conclusions

In conclusion, addressing anemia in Indonesia requires a comprehensive and multi-faceted approach. By implementing a comprehensive approach that combines education, counseling, ANC improvements, and fortified milk supplementation, it is possible to reduce anemia prevalence and improve maternal and child health outcomes in Indonesia. This approach aims to increase awareness, enhance knowledge, and provide the necessary support to ensure better overall health and well-being for the population at risk. However, while this review focuses primarily on anemia in breastfeeding women in Indonesia, the findings may not be broadly generalizable to other countries and contexts.

## Figures and Tables

**Figure 1 nutrients-16-01285-f001:**
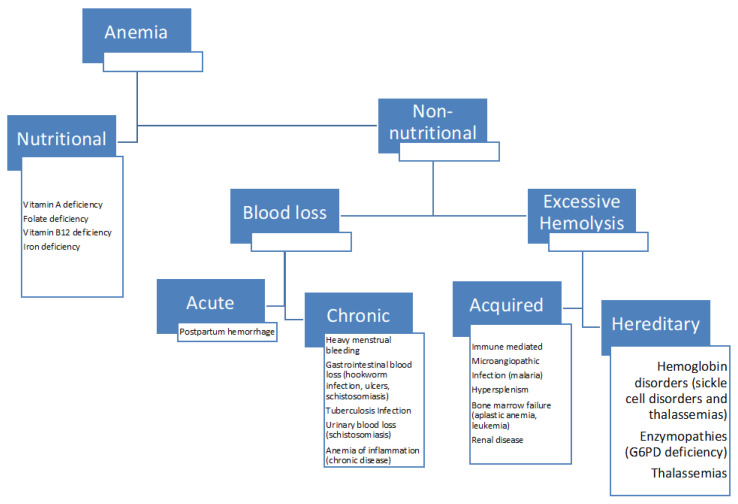
Common causes and classifications of anemia.

**Table 1 nutrients-16-01285-t001:** The iron concentration of breast milk from women of term infants aged 1–6 months according to country, from the literature (from the year 2000 onwards) adapted to Daniels et al., 2018 [16].

Country			mg/L	
Western Pacific (Japan)	1–3 months 3–6 months	1155 ^1^ 1155 ^1^	1.8 ± 3.3 0.5 ± 1.4	Yamawaki et al. (2005) [17]
Americas (Brazil)	2 months	31	0.9 ± 0.5	Mastroeni et al. (2006) [18]
South-East Asia (India)	3.5 months	100 ^3^	0.3	Shashiraj et al. (2006) [19]
Americas (USA)	1–1.5 months 2.5–3 months	30 17	0.5 ± 1.0 0.4 ± 0.3	Hannan et al. (2008) [20]
Eastern Mediterranean (Iran)	3–4 months	182	0.9 ± 0.5	Mahdavi et al. (2009) [21]
Western Pacific (Mongolia)	1–6 months	66	0.5 ± 0.2	Shi et al. (2011) [22]
Western Pacific (South Korea)	3 months 4 months 5 months	87 ^2^ 86 ^2^ 66 ^2^	0.4 ± 0.3 0.4 ± 0.7 0.3 ± 0.2	Kim et al. (2017) [23]
South-East Asia (Indonesia)	2–5.3 months	113	0.2 (0.2, 0.3)	Daniels et al. (2018) [16]

^1^ Sample included an unknown number of infants born <2.5 kg and/or mothers using supplements (unknown number and type). ^2^ 50% of the total sample took supplements (unknown type). ^3^ Included anemic and non-anemic women.

**Table 2 nutrients-16-01285-t002:** Summary of studies showing the association between maternal anemia and the outcomes in their breastfed child.

Study	Country	Child’s Age	Outcomes
Meinzen-Derr et al. (2006) [45]	Mexico	0–6 months old	Maternal anemia was significantly associated with their children’s anemia with odds ratio of 3.3 (95% CI = 1.1–9.9; *p* = 0.03).
Teixeira et al. (2010) [37]	Brazil	6 months old	Children receiving exclusive breastfeeding whose mothers had maternal anemia had significantly lower hemoglobin concentration compared to those whose mothers did not have maternal anemia (10.1 g/dL vs. 10.8 g/dL; *p* = 0.03).
Marques et al. (2014) [48]	Brazil	1–6 months old	Maternal iron deficiency and iron-deficiency anemia are not significantly associated with their infants’ iron deficiency and iron-deficiency anemia (*p* > 0.05).
Marques et al. (2016) [44]	Brazil	0–6 months old	Maternal Hb correlated significantly with their offspring’s Hb, especially at the age of 4 and 5 months old (β = 1.134 and 0.845, respectively).
Reinbott et al. (2016) [46]	Cambodia	3–23 months old	Maternal anemia was significantly associated with their children’s anemia at the age of 6–24 months old (*p* = 0.008). Maternal anemia increased the risk of their children’s anemia as high as 1.77 times in children aged 6–12 months old and 1.82 times in children aged 12–24 months old.
Rini et al. (2019) [42]	Indonesia	0–6 months old	Anemia during lactation had a significant low to moderate correlation with infant weight (r = 0.447 = *p* < 0.001).
Sudaryati et al. (2019) [41]	Indonesia	0–6 months old	Anemia in lactating mothers is not associated with infant growth (*p* = 0.161) and nutritional status (*p* = 0.685).

β = correlation coefficients; CI = confidence interval; Hb = hemoglobin.

## Data Availability

Not applicable.

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
