# Peer review of "Anemia in Breastfeeding Women and Its Impact on Offspring’s Health in Indonesia: A Narrative Review"

_nutrients, 2024, doi:10.3390/nu16091285_

Round 1

Reviewer 1 Report

Comments and Suggestions for Authors

Dear Authors,

   I have read your manuscript entitled “Anemia in Breastfeeding Women and Its impact of Offspring’s Health in Indonesia: A narrative Review” submitted to be considered for publication in Nutrients.

This work attempted an ambitious goal: to unravel all the potential contributors to anemia in lactating mothers from a region like Indonesia. The narrative review is useful to analyze and discuss such a Health Public Problem, however the manuscript shows some weaknesses that should be pointed out:

-            Once and again, it is referred to the serious consequences for both the mother and her child (lines 37, 39, 98, 180). It would be desirable to include a table that summarizes maternal as well as infant outcomes derived from anemia.

-            There are references that hardly match with the text, such as references 24 to 28. This is particularly striking in the case of reference 37.

-            But more importantly, the guiding thread of this narrative should be substantially improved to make the manuscript more readable. The section about management programs (pag 4) includes studies about impacts on children’s health (lines 131-138). The paragraph (lines 221-235) is not directly related to the consequences on infants, but more in consonance with sections 2 or 3.

Author Response

Please see the response for Reviewer 1 in the attached rebuttal letter

Reviewer 2 Report

Comments and Suggestions for Authors

Dear authors,

I have now completed the review of the manuscript titled "Anemia in Breastfeeding Women and Its Impact on Offspring’s Health in Indonesia: A Narrative Review."

In the present study, the authors provide a useful overview of an important public health problem. 

The manuscript is interesting and, in general, fairly well-written.

I have some suggestions to further improve the quality of the manuscript.

I would like to suggest that the authors address these limitations in the article, either by discussing them in the limitations section or, where feasible, by making the appropriate revisions:

1. The article focuses primarily on anemia in breastfeeding women in Indonesia. While it provides valuable insights for this specific population, the findings may not be broadly generalizable to other countries or contexts. The authors could discuss the potential limitations of the study's scope.

2. The authors missed the prior research during review - Breastfeeding and impact on childhood hospital admissions: a nationwide birth cohort in South Korea. This article could provide insights into potential long-term impacts of breastfeeding on child health outcomes, which is relevant given the focus on the effects of maternal anemia during breastfeeding on infants in the authors' article.

3. There are limited discussion of confounding factors. While the article identifies several risk factors and causes of anemia, it doesn't thoroughly explore potential confounding variables that could influence the observed associations between maternal anemia and child health outcomes. 

Socioeconomic status, education, and access to health services are mentioned but warrant deeper analysis. For example, it would be better to cite Associations between Delayed Introduction of Complementary Foods and Childhood Health Consequences in Exclusively Breastfed Children. This mentions the importance of the first 6 months of exclusive breastfeeding for infant iron status. This selected article may provide additional context on the timing of complementary food introduction and its impact on child health in breastfed infants. Also, authors should look into Association between complementary food introduction before age 4 months and body mass index at age 5-7 years: a retrospective population-based longitudinal cohort study. Similar to the previous article, this one focuses on the timing of complementary feeding in relation to a specific childhood health outcome (BMI). It could shed light on the importance of breastfeeding and complementary feeding practices.

4. The article describes the importance of iron supplementation but could provide more specifics on effective intervention strategies, dosages, and implementation challenges. More detailed recommendations for policies and programs would strengthen the article's impact. For example, there was an article examined associations between infant feeding practices and later disease outcomes in children - "Relationship between feeding to sleep during infancy and subsequent childhood disease burden." It could potentially include information relevant to the long-term impacts of breastfeeding and maternal nutrition on child health.

5. Last and minor, organizing the article around a clear conceptual framework illustrating the relationships between risk factors, anemia, and outcomes could enhance the article's clarity and theoretical contributions. This can be considered as a lack of conceptual framework.

Thank you for your valuable contributions to our field of research. I look forward to receiving the revised manuscript.

Author Response

Please see the response to Reviewer 2 in the attached rebuttal letter

Round 2

Reviewer 1 Report

Comments and Suggestions for Authors

The authors have addressed all my queries. 

No further comments